# Hidden in the Eyes—Recurrence of Systemic Hemopathies Reportedly “In Remission”: Six Cases and Review of Literature

**DOI:** 10.3390/medicina58030456

**Published:** 2022-03-21

**Authors:** Margot Denier, Sarah Tick, Romain Dubois, Remy Dulery, Andrew W. Eller, Felipe Suarez, Barbara Burroni, Claude-Alain Maurage, Claire Bories, Johanna Konopacki, Michel Puech, Didier Bouscary, Alberte Cantalloube, Emmanuel Héron, Ambroise Marçais, Christophe Habas, Vincent Theillac, Chafik Keilani, Gabrielle R. Bonhomme, Denise S. Gallagher, Julien Boumendil, Wajed Abarah, Neila Sedira, Stéphane Bertin, Sylvain Choquet, José-Alain Sahel, Lilia Merabet, Françoise Brignole-Baudouin, Marc Putterman, Marie-Hélène Errera

**Affiliations:** 1Centre Hospitalier National des Quinze-Vingts, CIC 1423, DHU Sight Restore, 28 rue de Charenton, Sorbonne-Universités, Université Pierre et Marie Curie (UPMC), 75012 Paris, France; m.deniergaudron@gmail.com (M.D.); stick@15-20.fr (S.T.); alberte.cantalloube@wanadoo.fr (A.C.); eheron@15-20.fr (E.H.); chabas@15-20.fr (C.H.); v.theillac@gmail.com (V.T.); ckeilani@15-20.fr (C.K.); j.boumendil@15-20.fr (J.B.); nsedira@15-20.fr (N.S.); sbertin@15-20.fr (S.B.); sahelja@upmc.edu (J.-A.S.); l.merabet@15-20.fr (L.M.); francoise.brignole-baudouin@inserm.fr (F.B.-B.); puttermanmarc@gmail.com (M.P.); 2Institut de Pathologie, CHRU de Lille, 59000 Lille, France; romain.dubois@chru-lille.fr; 3Service d’Hématologie Clinique et de Thérapie Cellulaire, Hôpital Saint Antoine, AP-HP, Sorbonne Université, INSERM UMRs938, 75012 Paris, France; remy.dulery@aphp.fr; 4Ophthalmology Service, University of Pittsburgh School of Medicine, Pittsburgh, PA 75012, USA; elleraw@upmc.edu (A.W.E.); bonhommegr@upmc.edu (G.R.B.); gallagherds@upmc.edu (D.S.G.); 5Department of Hematology, Hôpital Necker-Enfants Malades, INSERM UMR 1163 et CNRS ERL 8254, Institut Imagine, Sorbonne Paris Cité, Université Paris Descartes, 149 rue de Sèvres, CEDEX 15, 75743 Paris, France; felipe.suarez@aphp.fr (F.S.); ambroise.marcais@aphp.fr (A.M.); 6Centre de Recherche des Cordeliers, Sorbonne Université, Inserm, Université de Paris, 75006 Paris, France; barbara.burroni@aphp.fr; 7Centre de Recherche Jean-Pierre Aubert INSERM: U837, Université du Droit et de la Santé—Lille II, Faculté de Médecine 1, Place de Verdun, CEDEX, 59045 Lille, France; claude-alain.maurage@chru-lille.fr; 8Institut de Pathologie—CHRU de Lille, 59000 Lille, France; 9France Department of Hematology, CHRU, 59000 Lille, France; claire.bories@live.fr; 10Department of Hematology, Hôpital D’instruction des Armées Percy, 92140 Clamart, France; jokonopacki.hematopercy@gmail.com; 11Centre Explore Vision, 75001 Paris, France; drmichelpuech@gmail.com; 12Department of Hematology, Faculté de Médecine Sorbonne Paris Cité, Université Paris Descartes, Hôpital Cochin, AP-HP, 75014 Paris, France; didier.bouscary@aphp.fr; 13Department of Hematology, Hôpital de Meaux, 77100 Meaux, France; wabarah@ghef.fr; 14Department of Hematology, Hôpital Pitié-Salpêtrière, 75013 Paris, France; sylvain.choquet@aphp.fr; 15Sorbonne Universités, 75006 Paris, France

**Keywords:** acute lymphoblastic leukemia, acute myeloid leukemia, diffuse large B-cell lymphoma, eye neoplasms, multiple myeloma

## Abstract

*Background and Objectives*: Secondary ocular localizations of hematological malignancies are blinding conditions with a poor prognosis, and often result in a delay in the diagnosis. *Materials and Methods*: We describe a series of rare cases of ocular involvement in six patients with hematological malignancies, reportedly in remission, who presented secondary ocular localizations, challenging to diagnose. Two patients had an acute lymphoblastic leukemia (ALL) and developed either a posterior scleritis or a pseudo-panuveitis with ciliary process infiltration. One patient had iris plasmacytoma and developed an anterior uveitis as a secondary presentation. Two patients had a current systemic diffuse large B-cell lymphoma (DLBCL) and were referred either for intermediate uveitis or for papilledema and vitritis with secondary retinitis. Finally, one patient with an acute myeloid leukemia (AML) presented a conjunctival localization of a myeloid sarcoma. We herein summarize the current knowledge of ophthalmologic manifestations of extramedullary hematopathies. *Results*: Inflammatory signs were associated with symptomatic infiltrative lesions well displayed in either the iris, the retina, the choroid, or the cavernous sinus, from the admission of the patients in the ophthalmological department. These findings suggest that patients with ALL, AML, systemic DLBCL, and myeloma can present with ophthalmic involvement, even after having been reported as in remission following an effective systemic treatment and/or allograft. *Conclusions*: Early detection of hidden recurrence in the eyes may permit effective treatment. Furthermore, oncologists and ophthalmologists should be aware of those rare ocular malignant locations when monitoring patient’s progression after initial treatment, and close ophthalmologic examinations should be recommended when detecting patient’s ocular symptoms after treatment.

## 1. Introduction

Hematological malignancies develop from hematopoietic cells. The World Health Organization international consensus classification considers the original tissue of the proliferation (lymphoid or myeloid) and the clinical, morphological or histological, immunophenotyping, genetic, and molecular information to define each entity [1]. Malignant hemopathies are divided into leukemia, lymphoma, and myelodysplasic syndromes.

An estimated total of 176,200 people in the US were expected to be diagnosed with leukemia, lymphoma or myeloma in 2019 [2], and in France in 2012, 35,000 new cases of hematological malignancies were reported [3]. These hematological malignancies can provoke ocular and orbital localizations due to direct infiltrative pathways or secondary to the hematopoietic disorders (hyperviscosity, anemia, thrombopenia, lymphopenia) or, in some cases, be iatrogenic in origin.

Direct infiltrating involvement within the eye is rare, therefore, when it does occur, it is related to the accumulation of circulating leukemic cells in the uvea [4,5]. It can appear as leukemic pseudo-hypopyon or optic disc infiltration, such as a papilledema with peripapillary hemorrhages or white-grayish subretinal lesions due to choroidal infiltration. Other specific lesions include orbital infiltration (causing exophthalmos and/or palpable masses along the orbital border) and, less commonly, palpebral, conjunctival, and lacrimal gland infiltrations [4,5,6,7].

Orbital Multiple Myeloma (MM) may present as plasmacytoma, which is an isolated tumor of monoclonal plasma cells in the absence of other skeletal lesions. Orbital MM may also present as a primary/solitary extramedullary plasmacytoma that may invade the orbit by infiltration from the sinuses [8].

The ophthalmic manifestations of MM can be seen in practically every ocular structure. Ophthalmic manifestations include proptosis, diplopia, lid ecchymosis, xanthomatosis, conjunctival and corneal deposits, scleritis, episcleritis, secondary glaucoma, ciliary body cysts, ciliochoroidal effusion, uveal plasmacytoma, hyperviscosity retinopathy, retinal vasculitis, detachment of sensory retina and retinal pigment epithelium, and neuro-ophthalmic manifestations [9,10,11,12,13,14,15,16,17,18,19,20,21,22].

The designation of intraocular lymphoma includes primary intraocular lymphoma, mainly arising from the central nervous system (CNS) and secondary intraocular lymphoma from outside the CNS as metastasis from a non-ocular neoplasm [23,24]. Lymphoma can affect the vitreous, retina, optic nerve, and subretinal pigment epithelium in the form of the primary vitreoretinal lymphoma, mainly arising from CNS. These are predominantly of B-cell origin, however, there are also rare T-cell variants [25,26]. The first signs are typically bilateral myodesopsias in none red and none painful eyes, with sight maintained for months. The diagnosis can be made by vitreous biopsy.

Lymphoma can arise in the ocular adnexal (OA) compartments, i.e., the structures and tissues surrounding the eye, including the conjunctiva, the eyelids, the lacrimal gland, and orbital soft tissue [27]. Most lymphomas are low-grade B-cell lymphomas, with extranodal marginal zone B-cell lymphoma of MALT type (mucosa-associated lymphoid tissue) being the most common type, and a high frequency of follicular lymphoma [27]. Mantle cell lymphoma, chronic lymphocytic leukemia, lymphoplasmacytic lymphoma, and splenic marginal zone lymphoma less frequently affect the ocular adnexa, but rarely at initial presentation. This contrasts with cases of marginal zone lymphoma and follicular lymphoma primarily involving the OA [27,28,29]. The other types of lymphomas are rare: lymphoblastic lymphomas (of T-lineage and of precursor B-cell type), B-cell lymphoma, peripheral T-cell lymphoma, NK/T-cell lymphoma, classic Hodgkin lymphoma, Burkitt lymphoma, T-cell lymphoma, and NK-cell lymphoma present with OA involvement or involve the OA secondarily [29].

Acute myeloid leukemia (AML) is the most common acute leukemia in adults, accounting for ~80 percent of cases in this group. Within the United States, the incidence of AML ranges from three to five cases per 100,000 population [30]. The diagnosis of acute leukemia is established by the presence of 20% or more blasts in the bone marrow or peripheral blood [31]. Most of the clinical manifestations of AML reflect the accumulation of malignant, poorly differentiated myeloid cells within the bone marrow, peripheral blood, infrequently in other organs, and, rarely, in the eye [32]. In two prospective studies, none of the 116 patients with AML were found with secondary ocular infiltration [33,34]. The signs are infiltration of the choroid, the sclera, the episclera, the conjunctiva, and the optic nerve.

The authors herein describe six patients with rare secondary ocular locations of systemic hematological malignancies considered to be in remission: acute lymphoblastic leukemia (ALL), AML, systemic diffuse large B-cell lymphoma (DLBCL), and MM. Information about ophthalmic manifestations of systemic hematological malignancies was obtained from journal articles and review articles.

## 2. Materials and Methods

The medical records of all patients diagnosed with ocular localizations of systemic hematological malignancies at the Quinze-Vingts National Eye Hospital, Paris, France between 2010 and 2017 and in UPMC, Pittsburgh, PA, USA in 2018 and 2019 were reviewed. Inclusion criteria comprise ocular localization with inflammatory signs associated with symptomatic infiltrative lesions in either the iris, the retina, the choroid, or the cavernous sinus of a systemic hematological malignancy, reportedly in remission, and histopathologic confirmation of malignant cells in the eye.

Patient data analyzed in this study included age at diagnosis, gender, past medical history, best-corrected Snellen visual acuity (VA), affected eye, symptoms, referral diagnosis, and prior treatment.

Literature searches were performed using electronic medical databases on the ophthalmologic features of the following hematological malignancies: ALL, systemic DLBCL, and MM. PubMed served as the primary database for the electronic literature search. In addition, the bibliographies of existing literature reviews and key articles were reviewed to identify other relevant articles appropriate for inclusion. The goal of the literature search strategy was to identify published articles for which the topic of interest was the primary focus, rather than all articles on the topic. Internet searches provided supplemental information, thus ensuring that interpretation of the identified articles was consistent with current knowledge. We excluded studies that were not published in the English language and those that did not report research results related to the key question.

## 3. Results

### 3.1. Presentation of Our Cases

#### 3.1.1. Case 1. First Relapse of an ALL

A 51-year- old woman was referred for posterior scleritis. She had complained of decreasing vision, metamorphopsias, and a painful right eye for the previous two weeks. Her medical history was significant for acute T cell lymphoblastic leukemia (pre-T), non-hyperleucocytic, with normal karyotype and no molecular abnormalities, diagnosed four years ago. Initially, peripheral blood showed 66% of blasts cells and myelogram and 86% of CD45 positive blasts (Table 1).

Other positive markers were CD2/5/7 associated with CD8 expression without CD1a, therefore consistent with ALL pre-T/T (ALL-European Group of Immunological Markers for Leukemias scoring of TII/TIII). She was treated with chemotherapy induction, then consolidation (GRAALL-2005 protocol, standard arm). Full cytologic remission was obtained in the six months prior to presentation at the Ophthalmology clinic. The last hematologic work-up, performed 14 months after the end of therapeutic treatment, was reassuring. Her VA was reduced to hand motion in the right eye. Ophthalmic examination found relative afferent pupillary defect in the right eye. Fundus examination showed a papilledema with serous retinal detachment and a mild vitreous haze at the macula. Spectral domain optical coherence tomography (SD-OCT) confirmed the subretinal fluid at the fovea and further showed a wavy (retinal pigment epithelium) RPE contour that corresponded to choroidal folds and to a massive choroidal infiltration in the macular area. Fluorescein angiography showed early delayed perfusion of the involved macular area, followed later on by fluorescein hypofluorescence of the same macular area with leakage of the optic nerve (papillitis) (Figure 1A–D).

A recent brain magnetic resonance imaging (MRI) was normal. An ocular B-scan ultrasound showed a typical aspect of localized posterior scleritis, with posterior coat thickness measured at 3.7 mm in the right eye. SD-OCT showed choroidal folds and sub-retinal fluid (Figure 1E,F). Both findings were compatible with posterior scleritis. Since the infiltration may appear to explain the presented signs, the aqueous humor was sampled to look for malignant cells. The aqueous tap was normal with no suspicious cells (no blasts). Routine uveitis work-up for scleritis was negative and the hematology team was consulted; they confirmed that the most recent hematologic work-up was negative for malignancy. Intravenous corticosteroids were initiated to treat the scleritis, followed by a slow tapering without efficacy. A few weeks later, the patient complained of a worsening of vision. A periocular biopsy was performed because of intraconic fat infiltration in the retro-ocular region and histopathology showed cells infiltrating almost all the adipose tissue (Figure 1G–J). Those cells were of moderate dimension with a high nucleocytoplasmic ratio and a hyperchromatic, angular nucleus. Chromatin was condensed and contained multiple, small nucleoli. There was frequent mitosis. Immunohistochemistry found a positive staining CD3+ CD10+ (staining of the nucleus of all TDT and Ki67+ tumor cells). A positron emission tomography (PET)/computed tomography (CT) scan found hypermetabolic lesion in the liver, the thymus, the kidneys, pulmonary micronodules, and one retro-ocular lesion in the right eye. We concluded that orbital localization is a phenotype T cell lymphoblastic proliferation, as a manifestation of an extramedullary relapse of lymphoid T cell lymphoblastic leukemia. In this case, the ocular location led to salvage chemotherapy (as per COOPRALL protocol 2007) where L-asparaginase had been replaced by crisantaspase, and the patient received four triple intrathecal injections. An allogeneic hematopoietic stem cell transplantation was also programmed. Four months later, the appearance of a left facial paralysis associated with a deficit of the lower left limb led to evidence of a second relapse of ALL. A catch-up treatment with clofarabine and aracytin with cytarabine once daily and clofarabine once daily from D1 to D5 was initiated. The PET-Scan control was in favor of a partial response with persistence of foci of renal fixation. After a first consolidation chemotherapy, a control PET scan showed an increase in the volume and intensity of the kidney parenchyma, which was biopsied, showing a known localization of ALL. A lumbar puncture was performed that regained a blast invasion of the CSF. Unfortunately, this patient died after inefficacy of anti-NOTCH treatment.

#### 3.1.2. Case 2. Relapse in the Form of Iris Plasmacytoma

A 58-year-old woman was referred to the uveitis clinic with anterior uveitis in the left eye. Her medical history was significant for IgG lambda MM, stage 3, diagnosed four years previously (Table 1). She exhibited bone involvement, serum protein electrophoresis with an abnormal peak at 39 g/L, creatininemia at 103 µmol/L, anemia at 9.8 g/dL, beta2 microglobulinemia at 4.23 mg/L, and caryotype with deletion of 13. Initially, she was treated with chemotherapy (VAD then DCEP), then underwent an autograft procedure with maintenance treatment (lenalidomide); however, this resulted in an incomplete response. Due to a relapse of the disease process, another course of chemotherapy was followed by a second autograft, resulting in a full remission (Table 1). Ocular symptoms occurred 12 months after completion of the last chemotherapy. Her VA was reduced to hand motion in the left eye. Ophthalmic examination and SD OCT in the left eye showed keratic precipitates, predominantly inferiorly, and an iris mass (Figure 2A,B). A brain MRI showed a contrast enhanced (gadolinium uptake) process within the anterior chamber in the left eye (Figure 2C). Anatomic pathology from fine-needle aspiration biopsy of the iris with a 25-gauge needle going through a transcorneal route into the iris-corneal angle found cells with features similar to lymphoid cells, with cytological characteristics indicative of malignancy. Immunohistochemical reexamination was indicative of a lambda monotype (Figure 2D,E). The systemic work-up performed included a PET/CT scan which found two spinal bone lesions, which were treated with radiotherapy. Myelogram showed an excess of plasma cells (14%), dystrophic, but with present megacaryocytes, no dyserythropoiesis, and no dysgranulopoiesis. Immunophenotyping confirmed the monoclonal gammapathy: CD38+, CD138+, CD56+, CD19±, CD20±, CD17-.

Conventional radiotherapy of the iris (40 Gy) was commenced. Chemotherapy was re-commenced three months later because of an increasing peak of IgG lambda in the serum. Unfortunately, this patient died as a result of acute kidney insufficiency related to a kidney amyloidosis. In this case, the ocular disease was associated with a progression of the systemic myeloma.

#### 3.1.3. Case 3. Orbital Relapses in the Case of Systemic DLBCL, despite Auto and Allograft Transplant of Hematopoietic Stem Cells

A 68-year-old female was diagnosed with high-risk (IPI 3) disseminated DLBCL with lymph nodes, parasternal right mass, and bone marrow involvement (Table 1). Immunohistochemistry revealed that tumor cells were positive for CD20, BLC6, BLC2, and c-MYC and negative for CD10, MUM1, and CD5. The tumor cells had had a high proliferation (index 9Ki67 85%). Epstein-Barr virus encoded RNA in situ hybridization was negative. A rearrangement of BCL2, BCL6, and MYC (Dako split probes) genes was found by fluorescence in situ hybridization without rearrangement of BCL2 and MYC/IGH (Dako split and fusion probes). According to 2017 WHO classification, it was reclassified as high-grade B-cell lymphoma with rearrangement of MYC and BCL6.

Complete remission was achieved after four cycles of immune-chemotherapy with R-CHOP and methotrexate and followed by high dose chemotherapy with BEAM and autologous stem cell transplantation. Nine months after the initial diagnosis, the patient relapsed with posterior nasal scleral infiltration with lung and bone marrow involvement. Ophthalmic examination in the left eye showed areflexic mydriasis with a left ophthalmoplegia. Fundus examination showed optic nerve edema. Intraocular pressure was increased to 22 mmHg in the left eye. Her VA was reduced to 6/30 in the left eye (Snellen chart = 20/40, US chart). She was highly myopic and had an open angle glaucoma treated by topical antiglaucoma medication. Retinal angiographies and SD-OCT ruled out any active posterior inflammation in the left eye. B-scan echography found a posterior-nasal scleral thickening and MRI showed a left perineuritis (Figure 3A–C).

Second line therapy with R^2^-ICE was administered for four cycles, leading to a second complete remission (Table 1). During this second line of treatment, the patient presented with a VA reduced to 6/48 (20/60 Snellen chart = US chart) and an intermediate uveitis in the left eye. A cytokine biomarker of intraocular lymphoma, IL10, was tested in the aqueous humor after the results of an anterior chamber tap were found to be negative. The IL-6 level was increased to 41 pg/mL. A MRI of the brain performed one and three months later showed a full regression of the mass in the cavernous sinus, and the persistence of a mild infiltration of the apex of the left orbital fat and intraconic perineuritis. The intermediate uveitis became quiescent after periocular corticosteroid treatment.

The patient then underwent allogeneic stem-cell transplantation 19 months after initial diagnosis with a 9/10 HLA mismatched unrelated donor, following a reduced intensity conditioning (fludarabin, busulfan, and anti-thymocyte globulin). Graft-versus-host disease prophylaxis consisted of ciclosporin and short course methotrexate. Full hematological reconstitution was achieved by day 21. Grade I cutaneous acute graft-versus-host disease developed on day 30 and was treated with topical steroids. Cytomegalovirus reactivation occurred on day 30 and was treated with oral valaciclovir. A diagnosis of possible toxoplasmosis was made on the basis of a positive PCR for *Toxoplasma gondii* on day 30, and the patient was treated with sulfadiazine and pyrimethamine. Total body CT-scan and cranial MRI were normal.

Two months later, the VA dropped to 6/120 (Snellen chart = 20/400, US chart). Ophthalmic examination and B scan ultrasound showed cells in the anterior chamber and in vitreous with optic nerve oedema and an increased retinal thickness on SD-OCT (Figure 3D–F). The differentials were either lymphoma cells infiltration in a patient with history of systemic DLBCL or anterior/intermediate uveitis. A brain MRI showed a nodular thickening near the posterior coat of the left eye, continuous to the left optic nerve sheath that was thickened. This lesion was highly hypermetabolic on PET/CT scan and no other hypermetabolic lesions were detected. These data are consistent with a relapsed lymphoma post allogeneic hematopoietic cell transplantation.

A diagnostic standard 25 Gauge three-port pars plana vitrectomy was performed for flow cytometric immunophenotyping on vitreous biopsy samples, identifying one population of atypical lymphocytes larger than normal, CD20+, suspected of lymphoid proliferations (Figure 3G,H). IL-10 vitreous dosage was low (14 pg/mL) and IL-6 was elevated (637 pg/mL). Unfortunately, this patient died shortly after the diagnosis of relapsed lymphoma.

#### 3.1.4. Case 4. First Vitreous Relapse in the Case of Systemic DLBCL, despite CAR T-cell Therapy with Axicabtagene Ciloleucel. Second Retinal Relapse in the Form of a Retinitis after Pembrolizumab Infusions, Intravitreal Methotrexate, and Intrathecal Methotrexate

A 52-year-old female had a history of relapse refractory DLBCL, initially diagnosed 24 months prior (Table 1). She underwent the CAR T-cell therapy 12 months previously with associated papilledema in the left eye. The patient was also given one methotrexate prophylactically post CAR infusion due to previous CNS involvement. Three-month post axi-cel infusion, PET/SCAN showed complete metabolic resolution of disease. A complete assessment of her CNS, including a lumbar puncture with methotrexate and MRI of the brain, was unrevealing. She had an isolated recurrence within the left eye in the form of vitreous haze resembling a vitritis, with 4+cells, vitreous sheets, and diffuse pallor of the optic nerve. Flow cytometric immunophenotypic studies performed on the vitreous fluid demonstrated a population of monotypic B-lymphoid cells accounting for at least 20% of total events with the following immunophenotype: CD19 positive, CD20 positive, CD38 positive CD45 positive, kappa positive, and lambda negative (Table 1). Overall, these findings were consistent with involvement by B-lineage non-Hodgkin lymphoma with an immunophenotype that is similar to prior analyses. The diagnostic vitreous biopsy was obtained through pars plana vitrectomy. IL-10 measurement in aqueous humor was not available in that institution.

The patient was started on pembrolizumab infusions, intravitreal methotrexate injections weekly for three months and full evaluation of the central nervous system with intrathecal methotrexate. A good ocular response was noted because of vision improvement and significant reduction in vitreous haze and no chorioretinal infiltration. As regards her lymphoma, PET/CT showed no evidence of disease elsewhere. A repeat bone marrow biopsy was completed, which showed normal cytogenetics and no evidence of lymphoma. She was then developing a hemorrhagic retinal lesion in the left eye, resembling a retinal necrosis (Figure 4), that was revealing for progression of the disease one month after her last methotrexate intravitreal injection. An ocular viral involvement was ruled out by PCR in the aqueous humor, a repeated PET/CT showed two new fluorodeoxyglucose avid foci in the left anterior iliac wing corresponding to lytic lesions. A biopsy of the lesion confirmed a relapsed DLCL, with Ki67 score of 80%. PET 3/18/20 was negative, with the exception of left anterior iliac wing. A biopsy of the lesion confirmed relapse DLBCL with KI 67 of 80%. She resumed more weekly ocular injections and completed a treatment with external beam radiation therapy and polatuzumab/bendamustine/rituximab for six cycles. She underwent conditioning chemotherapy with fludarabine/melphalan and matched unrelated donor peripheral blood stem cell transplant conditioned with Fludarabine/Melphalan. Her day 100 marrow was hypocellular with trilineage hematopoiesis and no evidence of lymphoma. Cytogenetics were normal and PET/CT showed a complete metabolic response. On color fundus photography, there was a hyper/hypo pigmentation supra-nasally in an area of previous choroidal infiltrates that was now resolved after intravitreal methotrexate injections. The vision was stable.

#### 3.1.5. Case 5. Location in the Ciliary Process Associated with an Anterior Pseudo-Uveitis, Six Months after Completing a Treatment for ALL

A 30-year-old man presented to the Emergency Room in our institution with left anterior uveitis and associated pseudo-hypopyon. He was in complete remission after three lines of treatment for an ALL (Table 1). His last treatment was an allogeneic peripheral blood stem cell transplantation six months prior to presentation. He was in complete remission three months after the transplant with no signs of graft-versus-host disease and full donor chimerism. The cells obtained from the anterior chamber aspiration revealed blasts similar to the leukemic cells that were found at the time of the diagnosis of ALL (Figure 5A). An MRI showed a process and an infiltration at the level of the ciliary processes and an abnormal enhancement of the soft tissues in the superior extraconic space of the left orbit (Figure 5B). The bone marrow aspirate did not find any excess of blasts. The minimal residual disease by flow cytometric analysis was negative and there was no sign of CNS relapse. He received 8 Gy of radiotherapy and fully recovered from any presenting clinical symptoms. At last follow-up, at six months, the patient is alive and in complete remission.

#### 3.1.6. Case 6. Location in the Conjunctiva Associated with a Myeloid Sarcoma, Six Months after Completing a Treatment for ALL

A 25-year-old man presented to the Eye Casualty in our institution, referred by his hematologist, with red right eye with conjunctival infiltration (Figure 6A) since the last three days and normal fundus examination. He had been in complete remission of ALL for the last two years with pejorative prognostic factors.

His last treatments were azacitidine and sorafenib, followed by donor lymphocyte injections (Table 1) that were complicated by grade IV cutaneous, gastrointestinal, and hepatic corticoresistant graft-versus-host disease (GVH). GVH was treated by IV steroids infusion, anti-TNF alpha, alpha1-antitrypsine, mycophenolate mofetil, and ruxolitinib. The immunosuppressive treatment was discontinued 12 months prior to the current presentation with maintenance of photopheresis every 6 weeks.

The first conjunctival biopsy found a poorly differentiated tumor infiltration of a high grade, evoking a conjunctival localization of a myeloid sarcoma and suggesting a recurrence.

Since it is necessary to make molecular genetic tests, a second conjunctival biopsy was performed, and it found a positive IDH1 mutation and a duplication of the positive FLT3 gene.

A brain and orbit MRI showed a tumoral mucous thickening of the nasal cavity, turbinates, and left lacrimal gland and multiple bilateral cervical nodes, of identical signals to the mucous lesion. Its pathological appearance was in favor of a leukemic localization and an absence of intracranial abnormalities.

## 4. Discussion

We describe six patients with systemic hemopathies (ALL, DLBCL, and myeloma) that feature recent ophthalmic involvement, even after reported remission by effective systemic treatment and/or allograft. Early detection of hidden recurrences located in eyes is essential to permit effective treatment.

### 4.1. Cases 1, 5, and 6

In the current study, three cases revealed a secondary location of an ALL in the eye, which represents a very rare manifestation of ALL. Different studies have evaluated prospectively the prevalence of ocular localization in ALL, in adults and in children recently diagnosed, and before initiation of treatments. Reddy et al. found, among 288 patients, ocular localization in 35.4% of cases, but only 1.7% of infiltrative lesions [35]. In another series, Schachat et al. report that among 120 patients, there are abnormalities at examination in 39% of patients, 3% of these being leukemia infiltration. Reduced vision is observed in 5% of patients [4]. Karesh et al. found, among 53 patients presenting with recently diagnosed AML, that 34 of them had ocular involvement, but none had an infiltrative lesion [33].

Several reports have shown unusual and rare relapsing forms of acute leukemias where the only sign of relapse is anterior uveitis with pseudo-hypopyon, or a masquerade syndrome [7,36,37,38], the latter being a term used to describe disorders that simulate chronic uveitis. Case 5 of our study demonstrates this. When examining this patient, aspiration of intraocular cellular infiltrate for cytopathologic examination was required to obtain the correct diagnosis. A report by MacLean et al. showed blasts in the aqueous humor in three out of four children [7]; similarly, Novakovic et al. demonstrated blast infiltrations in seven out of nine children with similar clinical presentation [36]. The brain MRI in case 5 is informative because it shows the malignant process in the ciliary process associated with a felting of the fat tissues in the extraconic space (Figure 5B). Therefore, the secondary localization in the ciliary processes induced the hypopyon and the pseudo-uveitis features.

These case studies show early recurrences within the first months after ALL treatment, which is a high-risk period when specific maintenance treatments are tapered down [7,36,39]. Nonetheless, in case 1 of our study, the aqueous tap analysis did not show blastic cells; however, the absence of clinical flare in the anterior chamber in this case can explain the lack of malignant cells.

A periocular biopsy was performed due to severe infiltration of the fat in the intraconic space of the retro-ocular portion. The presence of tumoral cells supported the diagnosis of secondary localization of primary ALL in case 1 of our study (Figure 1G–J). A brain MRI was normal at the time of initial eye presentation, while a few weeks later, PET/CT scan found a hypermetabolic retro-ocular lesion in the index eye and elsewhere. It has been suggested that the eyes may remain a sanctuary site for leukemic cells during treatment, as they are routinely shielded during CNS irradiation, and many chemotherapy agents do not penetrate the eye wall [39]. It has been shown in pediatric populations that survival is greater in patients without orbital or ocular lesions. There was a higher frequency of leukemic relapse in bone marrow and/or in the CNS structures, in patients with specific orbital or ocular lesions compared to patients with nonspecific lesions, or in patients without lesions [6].

### 4.2. Case 2

Our second case documents an iris localization of a MM as a feature of relapsing plasmacytoma. The incidence of MM in the eye is difficult to assess. Postmortem histologic studies of 15 autopsied patients with MM showed intraocular lesions in 12 cases. Multiple pars plana cysts, retinal capillary microaneurysms, and two plasma cell lesions were identified. The plasma cell lesions took the form of a choroidal infiltrate in one patient and of a solitary plasmacytoma of the ciliary body in another patient [40]. In the first survey of international literature about orbital myelomatosis, Rodman and Font found 30 cases, including their 3 cases of orbital plasmocytomas with systemic manifestations of MM [41]. In 1987, De Smet and Rootman reported the diagnosis of three patients: one solitary plasmocytoma of bone, one known MM with plasmocytoma as an osteolytic mass in the orbit, and one disseminated MM with orbital infiltrate secondary to sinus involvement [42]. Later on, in a literature survey undertaken in 2009, Burkat et al. reported 41 articles in the English-language literature of 52 cases of orbital MM comprising of 61% of MM, 25% of plasmocytoma, 10% of primary extramedullary plasmacytoma, and 4% of necrobiotic xanthogranuloma. They reported that the time from diagnosis to orbital presentation was, on average, 17.6 months. In their series, 65% of patients already carried a diagnosis of MM, 35% of patients presented with an orbital manifestation having no prior diagnosis of MM, and 9.6% presented with initial signs of recurrent systemic disease [8].

In our case study, case 2 was diagnosed with IgG lambda MM stage 3, according to the International Staging System. Stage 3 is the most advanced stage, with a median overall survival of 29 months [10,43,44,45,46].

In case 2 of our study, the MM had been diagnosed four years previously, and the patient initially presented with bone involvement. At the time of the ocular involvement, the MM had been in complete remission for the previous 13 months, after two courses of chemotherapy and two bone marrow autografts related to a relapse.

Fine-needle aspiration biopsy of the iris lesion with anatomic pathology and immunohistochemical exams were necessary to identify malignant cells (Figure 2D,E).

Intraocular involvement in MM, as we showed in case 2, is considered rare. Burkat et al. found that bony involvement with local infiltration was the most common among the 52 cases reviewed [8].

### 4.3. Cases 3 and 4

In our study, the third case demonstrated a relapse within the eye after an initial systemic high-grade B-cell lymphoma with MYc and BCL6 rearrangements which was in complete remission as a result of allo and auto hematopoietic stem cell transplantations. Intraocular manifestations of systemic DLBCL are rare and there are no data about intraocular manifestations of high-grade B-cell lymphoma with MYc and BCL6 rearrangements. The fourth case also demonstrated a relapse within the eye after undergoing a CAR T-cell therapy and one methotrexate prophylactically post CAR infusion due to previous CNS involvement.

Indeed, AO of the systemic non-Hodgkin’s lymphoma occurs in roughly 5% of cases [47]. Moreover, in another study of 325 patients with systemic non-Hodgkin’s lymphoma [48], Lazzarino et al. reported that 2.4% of them presented with orbital tumors, whereas Bairey et al. [47] found that 5.3% of 187 such patients had orbital or OA involvement. Patients with primary adnexal lymphoma had a significantly better disease specific survival than those with systemic lymphoma [49]. Empirical research demonstrates that between 10% and 32% of all OA lymphomas are secondary [27,50,51,52].

Both cases 3 and 4 of our study required a diagnostic vitrectomy for the detection of malignant cells (Figure 3G,H, case 3), and lymphoid proliferation was identified by flow cytometric immunophenotyping. A brain MRI showed a nodular thickening near the posterior coat of the involved eye and a thickened optic nerve sheath (Figure 3A–C) and no other hypermetabolic lesions on the PET/CT scan. In case 4, PET/CT showed new lytic foci in the iliac wing secondary to active lymphoma.

### 4.4. General Discussion

Regarding the clinical aspect of the recurrence of ocular localizations of hematological malignancies in the eyes in our study, the difficulty in the diagnoses lay in the various features (or pseudouveitis) presenting in the context of a hemopathy in full remission: posterior scleritis and/or ciliary process, conjunctival and retroocular infiltrations in association with ALL, secondary plasmocytoma in the iris in association with MM, and choroidoscleral, orbital, and optic nerve infiltration in association with systemic lymphoma.

As regards the diagnostic tools (biology and imaging) used to perform the diagnosis of recurrence in the eye, we found that:

Cases 1, 5, and 6: The diagnosis of relapsing extramedullary manifestation of ALL in the fat of the intraconic space was made possible by biopsy only. The systemic work-up was negative in regard to possible relapses of ALL. Only the histology found the secondary ocular location and, later on, other secondary locations elsewhere were found by PET/CT scan a few weeks after the occurrence of posterior scleritis. In case 5, the diagnosis of a ciliary process secondary location of an ALL in remission was made based on the detection of blasts in the anterior chamber aspiration and MRI enhancement of the ocular malignant lesion. In case 6, the biopsy found a myeloid sarcoma, suggesting, in a patient on ALL in remission, an association with a tumoral infiltration of the nasal cavity, turbinates, and left lacrymal gland.

Case 2: The plasmocytoma was correctly diagnosed by iris biopsy in the context of MM in remission. The systemic work-up showed two spinal bone lesions by PET/CT scan.

Cases 3 and 4: In both cases, the vitreous biopsy with flow cytometric immunophenotyping was essential in identifying a lymphocyte population indicative of lymphoid proliferation.

In case 3, a brain MRI showed nodular thickening adjacent to the posterior wall of the left eye and contiguous to the left optic nerve sheath, thickened and highly hypermetabolic on PET/CT scan images. These findings are consistent with an early relapse of the lymphoma post allograft.

In case 4, PET/CT showed two new foci in the iliac wing corresponding to secondary lesions of lymphoma.

## 5. Conclusions

A combination of ocular biopsies (anterior chamber, vitreous, or orbital), depending on the location of the malignant ocular lesion and imaging of the eye by brain MRI, enabled the diagnosis in all cases. Analysis of ocular biopsies will obviously require accurate specific laboratory tests, which must be discussed with the relevant pathologist and hematologist, especially when volume of samples is small, i.e., aqueous humor. Our case series suggests that ocular manifestations must be investigated, even when a remission of the hemopathy is obtained. These hemopathies can present with a reservoir of disseminated tumor cells within the eye and represent a potential source of systemic relapse. All patients in our study received extensive specific treatment for their hemopathies.

Case 3 received auto and allo stem cells transplantation as a treatment for a systemic lymphoma, and case 5 received three treatment lines for an ALL and allogeneic peripheral blood stem cell transplantation. Case 4 received chemotherapy and CAR T cell therapy. Cases 1 and 2 received chemotherapy and/or conventional radiotherapy, extending their life duration, which allowed time for ocular and cerebral localizations to appear in the evolution of the disease. Case 6 received donor lymphocyte injections that were complicated by GVH, requiring an immunosuppressive treatment that might have favored the relapse of ALL.

We highlight the importance of performing recurrent eye examinations when monitoring such malignant hemopathy diseases (MM, ALL, etc.). Furthermore, hematologists and ophthalmologists should be aware of those rare ocular malignant locations when monitoring patients’ progression after initial treatment, or close ophthalmologic examinations should be recommended when detecting patients’ ocular symptoms after treatment.

### Clinical Practice Points

A combination of ocular biopsies (anterior chamber, vitreous, or orbital), depending on the location of the malignant ocular lesion and imaging of the eye by brain MRI, enabled the diagnosis in all cases. Analysis of ocular biopsies will obviously require accurate specific laboratory tests, which must be discussed with the relevant pathologist and hematologist, especially when volume of samples is small, i.e., aqueous humor.

Our case series suggests that ocular manifestations must be investigated, even when a remission of the hemopathy is obtained.

Hemopathies can present with a reservoir of disseminated tumor cells within the eye and represent a potential source of systemic relapse.

All patients in our study received extensive specific treatment for their hemopathies.

## Figures and Tables

**Figure 1 medicina-58-00456-f001:**
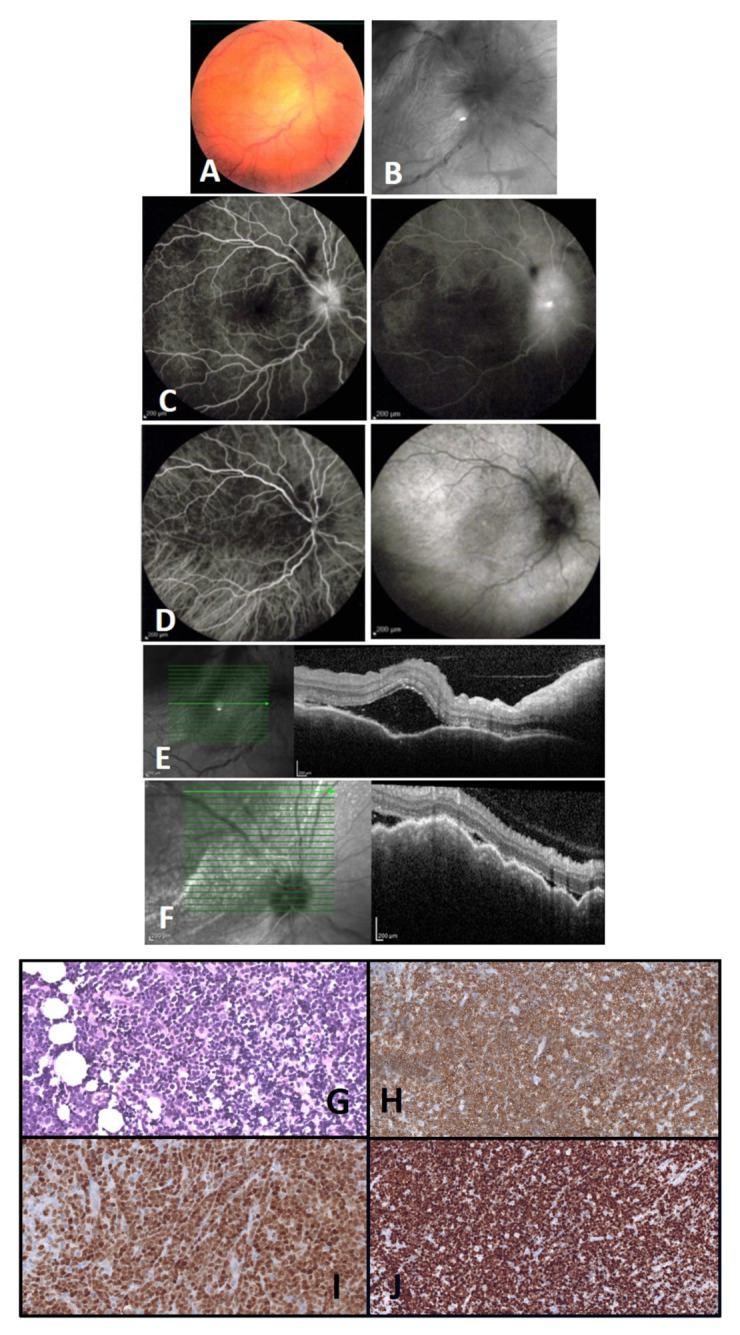
(**A**–**F**). Case 1. (**A**) Right eye color fundus (**B**) and red-free photographs showing a papilledema; (**C**) Early (on the left) and late (on the right) frame of fluorescein angiography showing dye leakage from the optic nerve; (**D**) Indocyanine green angiography early (on the left) and late (on the right) frames showing no obvious choroidal lesion; (**E**) Initial SD-OCT (Heidelberg) showing choroidal folds and localized subretinal fluid; (**F**) At two weeks, SD-OCT (Heidelberg) showing worsened choroidal folds and subretinal fluid. (**G**–**J**). Case 1. Histopathology and immunohistochemistry of a periocular biopsy; (**G**) Infiltration by lymphoblasts with round and irregular shaped nuclei (HES × 200); (**H**) Cd3 staining shows cytoplasmic positivity (CD3 clone NCL-L-CD3-565, ×200); (**I**) TdT staining highlights intense nuclear positivity (clone Novocastra NCL-L-TdT-339, ×200); (**J**) MIB1 antibody staining for Ki67 highlights very high proliferation rate (clone M7240, dako, ×200).

**Figure 2 medicina-58-00456-f002:**
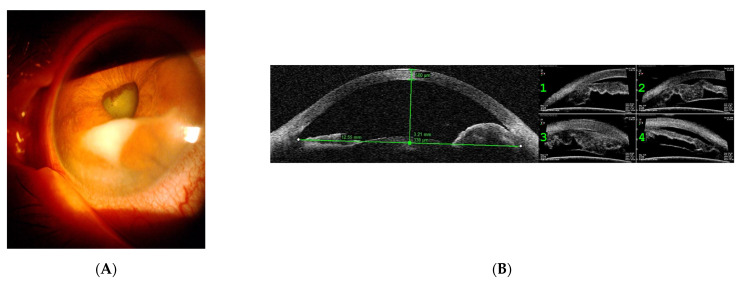
Case 2. (**A**) Slit-lamp photograph of the anterior segment of the left eye, showing anterior chamber involvement; (**B**) Ultrasound biomicroscopy of the left eye showing an iris mass with a tissular infiltrate at the base of the iris. (**C**) Brain IRM. Axial postcontrast fat-suppressed T1-weighted image (3T SIEMENS Skyra MRI scan) passing through the orbits and showing enhancement of the whole anterior chamber of the left eye. (**E**) Fine-needle aspiration biopsy of the lesion in the iris showing plasma cells producing IgG lambda (Immunohistochemistry) and (**D**) atypical plasma cells (May-Grünwald Giemsa).

**Figure 3 medicina-58-00456-f003:**
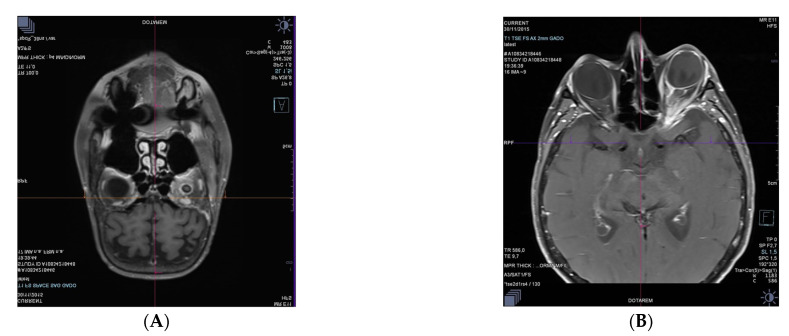
Case 3. Brain MRI. (**A**) Coronal postcontrast fat-suppressed T1-weighted image passing through the orbits and showing rostral peri- and intra-optic nerve enhancement on the left side; (**B**) Axial postcontrast fat-suppressed T1-weighted image passing through the orbits and showing choroidoscleral and perioptic meningeal enhancements extending to the adjacent orbital fat on the left side; (**C**) Oblique sagittal postcontrast fat-suppressed T1-weighted image showing the whole left optic nerve, whose meninges are strongly enhanced in the intral conal space (on the right) compared with the right optic nerve (on the left); (**D**) SD-OCT (Heidelberg) showing thickened retina (inner and outer retina); (**E**) SD-OCT (Heidelberg) showing foveal intraretinal cysts and subretinal fluid; (**F**) B-scan ultrasonography revealed left optic nerve thickening in shell appearance in two parts; one focal hyper-reflective part (white arrow) surrounded by a hypo-reflective, exudative area (white arrow head). Vitreous sample; (**G**) Atypical lymphocytes, large cells with large vesicular nuclei, nucleolated with basophilic cytoplasma (May-Grünwald Giemsa, 400×); (**H**) The lymphoid cells are identified as CD20+ (CD20, 400×).

**Figure 4 medicina-58-00456-f004:**
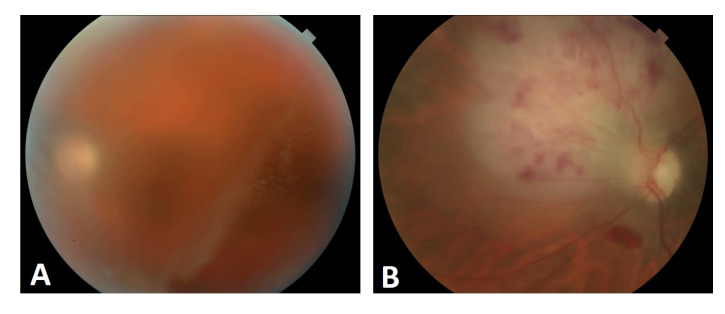
Fundus pictures, left eye; (**A**) showing vitritis before diagnostic vitrectomy; (**B**) showing infiltration of lymphoma cells into the retina causing a whitening of the retina nasally from optic nerve after vitreous biopsy.

**Figure 5 medicina-58-00456-f005:**
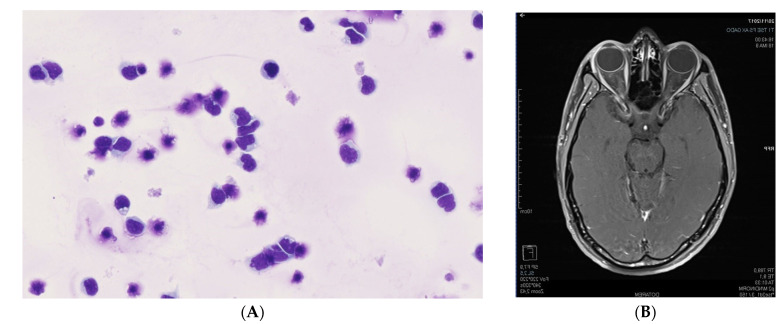
Case 5. (**A**) Anterior chamber aspiration; High-power view shows the blast cells in the anterior aqueous humour sample (May-Grünwald-Giemsa staining); (**B**) Brain MRI; Axial postcontrast fat-suppressed T1-weighted image passing through the orbits and the brain, showing enhancement of the ciliary processes of the left eyeball associated with abnormal enhancements of the dorsal pre-and extraconal post-septal spaces.

**Figure 6 medicina-58-00456-f006:**
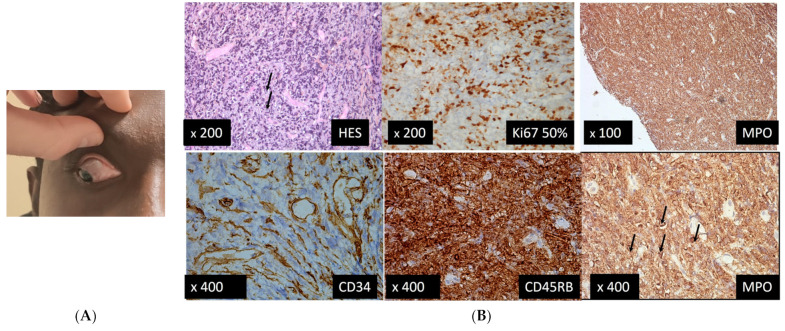
Case 6. (**A**) Feature of lymphoid infiltration of the nasal and superonasal conjunctiva in right eye; (**B**) Three out of 4 fragments of 5 × 8 mm that were analyzed revealed a diffuse high-grade tumor infiltration by blast cells, some of them with apparent nucleoli. The immunostainings found positive markers CD45RB, CD68KP, MPO and Ki67 on 50% of the cells, and negative markers CD34-, CD3- CD20-. The arrows indicate some blast cells.

**Table 1 medicina-58-00456-t001:** Showing the 6 cases and their corresponding diagnoses, immune phenotypes, and treatments.

	Diagnosis	Immune Phenotype	Treatments before Ocular Malignant Location
Case 1		Pre-T cell leukemia/T commune (EGIL scoring of TII/TIII). At presentation before ocular involvement CD33,65,61,64,14,MPO Negative B lymphoid markers: CD19,22,79aic Negative CD56. Periocular biopsy: lymphoblasts positive for TdT, CD10, and express CD3 T lineage specific marker.	Chemotherapy GRAALL-2005 protocol (46). Induction: prophase comprising 1 mg/kg/d oral PDN and one IT-MTX, then VCR, DNR, CPM, PDN, L-Aspa; Triple IT (from day 1–day 14), then (from day 15) VCR, DNR, CPM, and L-Aspa, G-CSF (from D18). Consolidation with MRX, aracytine, and cranial irradiation.
			Chemotherapy GRAALL-2005 protocol (46). Induction: prophase comprising 1 mg/kg/d oral PDN and one IT-MTX, then VCR, DNR, CPM, PDN, L-Aspa; Triple IT (from day 1–day 14), then (from day 15) VCR, DNR, CPM, and L-Aspa, G-CSF (from D18). Consolidation with MRX, aracytine, and cranial irradiation.
			Chemotherapy GRAALL-2005 protocol (46). Induction: prophase comprising 1 mg/kg/d oral PDN and one IT-MTX, then VCR, DNR, CPM, PDN, L-Aspa; Triple IT (from day 1–day 14), then (from day 15) VCR, DNR, CPM, and L-Aspa, G-CSF (from D18). Consolidation with MRX, aracytine and cranial irradiation.
Case 2	IgG lambda multiple myeloma	monoclonal gammapathy: CD38+ CD56+ CD19± CD20± CD17−	Initial chemotherapy (VAD followed by DCEP) then autograft procedure. - ocular relapse: another course of chemotherapy (bortezomib/lenalidomide/cyclophosphamide), then second autograft. Lenalidomide/dexamethasone, then melphalan/dexamethasone after radiotherapy of the iris was commenced. Bendamustine/bortezomib was re-commenced 3 months later because of an increasing peak of IgG lamba in the serum.
Case 3	HGBL with rearrangements of MYC and BCL6	Tumor cells were CD20(+), BCL6(+), BCL2(+), c-MYC+ CD10(–), MUM1(–), CD5(–), EBV(–) with Ki67 index at 85%. Rearrangements of BCL6 and MYC by FISH.	4 cycles of immune-chemotherapy with R-CHOP (rituximab/CPM/doxorubicin/VCR/PDN) and MTX followed by high dose chemotherapy with BEAM (BCNU/etoposide/cytarabine/melphalan). R-CHOP (rituximab, cyclophosphamide, doxorubicin, vincristine, prednisone) and methotrexate and then BEAM (BCNU, etoposide, cytarabine, melphalan). Second line therapy with R^2^-ICE (rituximab, ifosphamide, carboplatin, etoposide, and lenalidomide).
Case 4	diffuse large B-cell NHL, R-IPI score 2	At diagnosis: Bone marrow biopsy: hypercellular marrow (100%) with 90% involvement by CD20 positive large B-cell lymphoma. FISH for MYC, BCL2, and BCL-6 was negative.	data
	Diagnostic imaging showed splenomegaly, lymphadenopathy on both sides of her diaphragm and pancytopenia. At diagnosis, she had an associated HLH as her presentation included fevers, splenomegaly, cytopenias, ferritin of 13,230, elevated triglycerides of 972, fibrinogen of 175, and an elevated soluble IL2 receptor level of 3903	Vitrectomy biopsy: CD5 negative, CD10 negative, CD19 positive (variable/possibly partial), CD20 positive (variable/possibly partial), CD38 positive (moderate intensity), CD45 positive (slightly dim), kappa positive, and lambda negative. Also present are CD5 positive T cells.	- Initial chemotherapy: Pt received 2 cycles of R-CHOEP. Salvage therapy with ESHAP ×2 Bridging chemotherapy with gemcitabine/oxaliplatin 3x cycles CAR T cell therapy. Lymphodepleting chemotherapy with fludarabine and cyclophosphamide - First ocular relapse (vitreous involvement): Pembrolizumab infusions. Weekly intravitreal methotrexate, intrathecal methotrexate. - Second relapse (retinal involvement): Localized XRT Polatuzumab and BR (3 cycles)
Case 5	ALL	pro-T-cell ALL CD2- surface CD3 intra CD3+ CD5- CD7+ CD19- CD33+ CD34+ CD123+ intra MPO+	1st line: chemotherapy according to FRALLE protocol 2nd line: nelarabine 3rd line: induction chemotherapy with idarubicine and cytarabine followed by an allogeneic peripheral blood stem cell transplantation with sequential conditioning regimen from a matched unrelated donor
Case 6	ALL	poorly differentiated tumor infiltration of high-grade evoking a conjunctival localization of a myeloid sarcoma. Molecular biology: positive IDH1 mutation + duplication of the positive FLT3 gene	- pheno-identical allograft with myelo ablative conditioning therapy (fludarabine, busulfan) - 3 months later: faced with the reappearance of molecular markers of the disease and markers of the recipient at the level of chimerism, rapid decrease in immunosuppressive treatment - at 8 months, azacitidine + so-rafenib - donor lymphocytes injections that were complicated by GVH. GVH was treated by IV ster-oids infusion, anti-TNF alpha, alpha1-antitrypsine, mycophe-nolate mofetil, ruxolitinib with maintenance of photopheresis every 6 weeks.

A = doxorubicin (Adriamycin); ALL: acute lymphoblastic leukemia; ASCT: autologous stem cell transplantation; Aspa: L-Asparaginase; B: bendamustine; BCNU: Carmustine; C or CPM: Cyclophosphamide; DCEP (dexamethasone/ cyclophosphamide/ etoposide/ cisplatin); DNR: Daunorubicin; D: dexamethasone; E: etoposide; EGIL: European Group of Immunological Markers for Leukemias; GVH: graft-versus-host disease; HA: high dose cytarabine; IT-MTX: intrathecal injection of methotrexate; P: cisplatin (Platinol); PDN: prednisone; R: rituximab; S—solu-medrone; V or VCR: Vincristine; VAD: vincristine/doxorubicin/dexamethasone; XRT: Radiation Therapy.

## Data Availability

Not applicable.

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
