# Peer review of "Hidden in the Eyes—Recurrence of Systemic Hemopathies Reportedly “In Remission”: Six Cases and Review of Literature"

_medicina, 2022, doi:10.3390/medicina58030456_

Round 1
Reviewer 1 Report
- Very good case series shedding light on orbital and ocular manifestations of systemic malignancies.
- Great photos of ancillary studies and histology presented.
- Line 249: may delete the word "an"
- Line 261: I'm unfamiliar with the term, "tissular process"
- Line 264: instead of the the actual date, it can be helpful for the reader if you list the timeframe.
- Line 332: Is "eye Casualty" a term for clinic or emergency department?
- Line 339: I'm unfamiliar with the term, "felting"
- Line 358: should this say plasmapheresis?
- Line 361: "make molecular biology" does not make sense.
- The sentence starting in line 364 needs to be reworded to make sense.
- Line 469: "an" should be changed to "a"
- The Clinical Practice Points is a good addition
- Overall wonderful paper with great detail. This is an important topic
Author Response
- Very good case series shedding light on orbital and ocular manifestations of systemic malignancies.
- Great photos of ancillary studies and histology presented.
- Line 249: may delete the word "an". Thank you we amended accordingly.
- Line 261: I'm unfamiliar with the term, "tissular process". We agree and we have modified for “mass”
- Line 264: instead of the the actual date, it can be helpful for the reader if you list the timeframe. We made the changes accordingly.
- Line 332: Is "eye Casualty" a term for clinic or emergency department?. We modified for Emergency Room.
- Line 339: I'm unfamiliar with the term, "felting". ). We agree, and we modified for : “An MRI showed a process and an infiltration at the level of the ciliary processes”.
- Line 358: should this say plasmapheresis? It was a photophoresis treatment.
- Line 361: "make molecular biology" does not make sense. We modified for “molecular genetic tests”.
- The sentence starting in line 364 needs to be reworded to make sense. Thank you, we amended it as follows :” A brain and orbit MRI showed a tumoral mucous thickening of nasal cavity, tur-binates, and left lacrymal gland and multiple bilateral cervical nodes, of identical signal to the mucous lesion. Its pathological appearance was in favor of a leukemic localization and an absence of intracranial abnormalities.
- Line 469: "an" should be changed to "a". We made the change accordingly.
- The Clinical Practice Points is a good addition.
- Overall wonderful paper with great detail. This is an important topic . Thank you very much, your comments are much appreciated.
Reviewer 2 Report
Thank you for the opportunity to read this manuscript.
These are several clinical cases reported with a beautiful illustration. It raises a difficult problem which is the diagnosis of hemopathies with eye localization.
An overall remark would be that it seems surprising to mix both exceptional complications such as the localizations of acute leukemias and less rare localizations that are the localizations of systemic lymphomas.
It would be useful to have details on the practical modalities of the biopsy carried out with the fine needle of the iris nodule (clinical case 2).
For many clinical cases the therapeutic decision following the diagnosis of ophthalmological localization has not been indicated. It would be useful to precise this because it is sometimes difficult to adapt treatment at that time. In addition, it would also be interesting to have the follow-up of these patients to clarify the prognosis of these patients.
Author Response
These are several clinical cases reported with a beautiful illustration. It raises a difficult problem which is the diagnosis of hemopathies with eye localization.
An overall remark would be that it seems surprising to mix both exceptional complications such as the localizations of acute leukemias and less rare localizations that are the localizations of systemic lymphomas.
We agree.
It should be useful to have details on the practical modalities of the biopsy carried out with the fine needle of the iris nodule (clinical case 2).
Thank you for your comment, we added :” with a 25 gauge needle going through a transcorneal route into the iris‐corneal angle”.
For many clinical cases the therapeutic decision following the diagnosis of ophthalmological localization has not been indicated. It would be useful to precise this because it is sometimes difficult to adapt treatment at that time.
Thank you for your comments. We have added the therapeutic decisions in the manuscript.
For case 1, we added: “Intravenous corticosteroids were initiated to treat the scleritis, followed by a slow ta-pering without efficacy.”
For case 2, we added: “Conventional radiotherapy of the iris (40Gy) was commenced. Chemotherapy was re-commenced 3 months later because of an increasing peak of IgG lamba in the serum.”
For case 4, we added:“The patient was treated with a series of intravitreal methotrexate injections”.
In addition, it would also be interesting to have the follow-up of these patients to clarify the prognosis of these patients.
We are thankful for your comments, we have added the follow-up.
For case 1, we added: “In this case, the ocular location led to salvage chemotherapy (as per COOPRALL pro-tocol 2007).”
For case 2, we added: “Unfortunately, this patient died as a result of acute kidney insufficiency related to a kidney amyloidosis. In this case, the ocular disease was associated with a progression of the systemic myeloma.”
For case 3, we added : “Unfortunately, this patient died shortly after the diagnosis of relapsed lymphoma.”
For case 4, we added: “The patient is now in remission.”
For case 5, we added : “He received 8 Gy of radiotherapy and fully recovered from any presenting clinical symptoms. At last follow-up, at 6 months, the patient is alive and in complete remission.”